# Long-Term Effects of a Video-Based Smartphone App (“VIDEA Bewegt”) to Increase the Physical Activity of German Adults: A Single-Armed Observational Follow-Up Study

**DOI:** 10.3390/nu13124215

**Published:** 2021-11-24

**Authors:** Gesine Reinhardt, Patrick Timpel, Peter E. H. Schwarz, Lorenz Harst

**Affiliations:** 1Department for Prevention and Care of Diabetes, Faculty of Medicine Carl Gustav Carus, Technische Universität Dresden, Fetscherstrasse 74, 01307 Dresden, Germany; patrick.timpel@tu-dresden.de (P.T.); peter.schwarz@uniklinikum-dresden.de (P.E.H.S.); 2Department of Medicine III, Faculty of Medicine Carl Gustav Carus, Technische Universität Dresden, Fetscherstrasse 74, 01307 Dresden, Germany; 3Center for Evidence-Based Healthcare, Faculty of Medicine Carl Gustav Carus and University Hospital, Technische Universität Dresden, Fetscherstrasse 74, 01307 Dresden, Germany; lorenz.harst@tu-dresden.de; 4Paul Langerhans Institute, Faculty of Medicine, Technische Universität Dresden, Tatzberg 47, 01307 Dresden, Germany; 5German Center for Diabetes Research (DZD), Ingolstädter Landstraße 1, 85764 Neuherberg, Germany

**Keywords:** physical activity, app, prevention, long-term effect, sustainability

## Abstract

As physical inactivity is one of the four leading risk factors for mortality, it should be intensively treated. Therefore, this one-year follow-up study aimed to evaluate the long-term effects of a preventive app to increase physical activity in German adults under real-life circumstances. Data collection took place from July 2019 to July 2021 and included six online questionnaires. Physical activity was studied as the primary outcome based on MET-minutes per week (metabolic equivalent). Secondary outcomes included health-related quality of life based on a mental (MCS) and physical health component summary score (PCS). At the time of publication, 46/65 participants completed the study (median 52 years, 81.5% women). A significant increase of physical activity was observed in people with a low/moderate baseline activity during the first four months of follow-up (median increase by 490 MET-minutes per week, *p* < 0.001, r = 0.649). Both MCS (median increase by 2.8, *p* = 0.006, r = 0.344) and PCS (median increase by 2.6, *p* < 0.001, r = 0.521) significantly increased during the first two months and the BMI significantly decreased during the first six months after the intervention (median decrease by 0.96 kg/m^2^, *p* < 0.001, r = 0.465). Thus, this study provides evidence for the medium-term impact of the app, since the effects decreased over time. However, due to the chosen study design and a sizeable loss to follow-up, the validity of these findings is limited.

## 1. Introduction

Physical inactivity is a crucial risk factor for non-communicable diseases such as diabetes, coronary heart disease, and several types of cancer [1] and is therefore listed as one of the four leading risk factors for mortality by the World Health Organization [2]. Due to the linear relationship between physical activity and health status [3] and further positive effects on quality of life and mental health [4], physical activity should be promoted in all age groups [1,3,5].

Health apps have the potential to reach a large target group at low costs [6,7], for therapeutic as well as preventive purposes [8]. Fitness apps are especially popular and therefore potentially useful for the prevention of non-communicable diseases [9,10]. There is modest evidence supporting the effectiveness of health apps promoting physical activity [11,12], with interventions being more effective when containing evidence based content [13] in order to support and maintain behavior change (e.g., to promote motivation, reduce stress, and solve problems) [14]. However, despite the growing number of studies investigating health apps, only limited conclusions can be drawn about their full preventive potential for several reasons. First of all, most health apps promoting physical activity contain neither evidence-based content nor theory-based behavior change strategies [14,15], which reduces the likelihood of their effectiveness [13]. Additionally, most apps studied are not commercially available [16,17], and were predominantly tested in controlled clinical settings [6]. Both of these factors have led to a lack of information about real-world effectiveness. Furthermore, the validity of studies conducted in the past was often limited due to small sample sizes and short periods of data collection [18], leading solely to short-term effects [19]. The latter, in particular, is a considerable problem, since the prevention of non-communicable diseases requires sustained behavior change, which, according to the health behavior change model “Health Action Process Approach” (HAPA) by Schwarzer [20], consists of two phases (motivation and action), the completing of which requires time. One potentially effective strategy to deliver sustained behavior change and, thus, provide guide practice as well as reinforcement over time is the use of videos as visual stimuli [21,22,23].

Still, the full potential of preventive health apps remains unknown at present, resulting in a need for long-term studies investigating the effects of video-based interventions in a real-world setting. Therefore, the purpose of this study was to evaluate the long-term effects of an app-based eight-week prevention program (“VIDEA bewegt”) promoting physical activity in everyday life. It is part of the overall evaluation of the app “VIDEA bewegt”, consisting of a short-term study and this follow-up study [24]. Findings on the short-term effects of the app already confirmed significant changes in MET-minutes per week (metabolic equivalent) and health related quality of life directly after the intervention for those participants having completed the program [25]. This follow-up study focused on the first year after completion of the app.

## 2. Materials and Methods

### 2.1. Study Design

This study was designed as a single-armed observational study under real life conditions, designed to assess the long-term effect of the app in the first year after completion of the program. Data collection took place from July 2019 to July 2021, based on online questionnaires. A personal contact was not deemed necessary, considering the scope of the study. A detailed description of the rationale and the overall study design following the recommendations formulated by Eysenbach and the CONSORT-EHEALTH Group [26] has been published elsewhere [24].

In order to control for effects of health behavior not related to “VIDEA bewegt”, the usage of other health apps or participation in any health courses other than the ones provided by the app was included into the questionnaires.

This study was approved by the Ethics Committee of the Technical University Dresden (EK 272062019, 25 May 2019) and was registered in the German Clinical Trials Register (DRKS00017392, 14 June 2019).

### 2.2. Intervention

The app “VIDEA bewegt” is a certified, digital preventive program which aims to increase the physical activity of its users in everyday life using videos combining educational content and training instructions. Furthermore, the app contains several additional components to develop and maintain motivation and behavior change such as goal setting, documentation of progress, personal messages, and a chat function.

The app components are deeply rooted within theories of behavior change as well as existing evidence on successful behavior change strategies. Goal setting is an essential precondition for manifest behavior change according to the HAPA [20] and has been proven to be essential in the formation of a feeling of self-efficacy [27]. The same is true for any option to autonomously control one’s behavior [28], which is operationalized in “VIDEA bewegt” by including a tracking function for physical activity. Personal messages are essential for the individualization of behavior change strategies, which is a precondition for effective interventions [17,29] The chat function of “VIDEA bewegt” also enables social support, which is an important predictor of behavior change according to the ‘Theory of Planned Behavior’, wherein it is dubbed subjective norm [30]. All in all, the components of “VIDEA bewegt” are intended to enable participants to engage in active behavior change on their own free will (i.e., they trigger intrinsic motivation) [31]. The app therefore serves as a nudging devise promoting intrinsic motivation [32].

Furthermore, “VIDEA bewegt” was developed by an expert panel consisting of doctors, psychologists, sports scientists, nutrition specialists, and app designers, fulfilling the known success factor of an interdisciplinary development team [33].

The program is divided into eight course weeks, following a standardized structure, which can be completed by the users at individual pace. A face-to-face contact is not part of the intervention, but users have the possibility to contact experts in preventative health care and sports at any time via a chat function or connect with other users via a forum.

Apart from the nudging character of the app, intending to enable active and self-guided behavior change, the novelty of “VIDEA bewegt” is the mode of information transmission via video. Using narrative videos with relatable characters and a consistent story line is known to generate transportation effects which, in turn, reduce reactance towards behavior change strategies [34]. Video content tailored to individual preferences is especially effective in that matter [35], which is why the videos within “VIDEA bewegt” can be consumed at any time and as many times as desired. Furthermore, there are several additional videos available on various topics such as sports exercises that use different equipment and medical fun facts about digestion, metabolic types, and diet coke.

The app has been available on the German market for Android and iOS since March 2019. As a certified intervention, its costs are partially covered by statutory health insurance companies. As such it, is a pilot project in Germany since it is the first app of its kind that can be prescribed by a physician. For a detailed description of the structure and process of the intervention and the app components please see Appendix A. Further information about the intervention can be found on the German website of “VIDEA bewegt” [36].

### 2.3. Participants

The app’s target population was patients at risk for chronic disease, specifically middle-aged or older people with low levels of daily physical activity. The conditions for participating in the app-based program were being of legal age (≥18 years old) and the absence of serious medical conditions such as heart failure. There were no further restrictions to intervention use and all users were invited to take part in the evaluation. However, only study participants who successfully completed the whole intervention could participate in the follow-up study.

### 2.4. Sample Size

An a priori power analysis was conducted for the primary outcomes of the overall evaluation of “VIDEA bewegt” (intended to apply to the short-term and long-term study alike) predicting a minimum sample size of 27 participants [24]. However, the number of possible participants was not limited, and a larger sample size was aimed for in order to allow for additional (e.g., subgroup) analyses.

### 2.5. Preliminary Evaluative Measures

Before conducting the evaluation, the items of the questionnaire were validated by means of a think-aloud test (*n* = 7) and expert opinions. Furthermore, a pre-test of the online questionnaire was carried out with 21 individuals from the target group in order to assess the comprehensibility of the items and the technical performance of the questionnaire. Based on the findings, the wording of several questions was optimized and minor errors in the design of the survey were corrected. Apart from that, a usability test of the “VIDEA bewegt” application was conducted (*n* = 10) to gain a better understanding of the strengths and weaknesses of the app’s layout. The findings of this test were used to develop an optimized version of the app, which was used in this study.

### 2.6. Procedure

At the beginning of the intervention, all users were informed about the evaluation and asked whether they agreed to receive an email with further information about the study and access to the first questionnaire (B0). The completion of the first questionnaire, which included a consent form and privacy policy, was considered as consent to participate in the study. Invitations to further online questionnaires were sent out after completion of the program (F0) as well as two (F2), four (F4), six (F6), and twelve (F12) months later (see Figure 1).

Data collection was solely carried out based on the questionnaires and, therefore, all data was self-reported. No study visit occurred at any time during the intervention or follow-up period and no researcher was present when the questionnaires were completed.

### 2.7. Primary and Secondary Outcomes

Physical activity was the primary outcome of this study and measured as MET-minutes per week (metabolic equivalent, [37]) which were assessed with the Global Physical Activity Questionnaire (GPAQ) [38,39].

The secondary outcome of this study was the health related quality of life based on the Physical Component Summary Score (PCS) and the Mental Component Summary Score (MCS), which were obtained using the Short Form Health Survey (SF-8) [40,41].

Further outcomes included the BMI and body weight, physically active minutes per week in the domains of leisure, work, and transport, as well as sedentary hours per day. In addition to the measurements listed in the previously published study protocol [24], a more thorough description of the measurements used is given in the Appendix B.

### 2.8. Statistics

Sociodemographic data and sustained use were analyzed descriptively. Since the Shapiro-Wilk-Test confirmed the absence of a normal distribution for most of the primary and secondary outcomes (*p* at B0: weight = 0.026, BMI = 0.004, MET-minutes per week < 0.001, active minutes within the domains work/transport/leisure < 0.001, PCS = 0.123, MCS = 0.004), the Wilcoxon signed-rank test for dependent samples was used to test for significant changes in the main outcomes between the measurement points F0, F2, F4, F6 and F12 compared to baseline (B0). For the main outcome MET-minutes per week the Wilcoxon signed-rank test was also carried out to test for significant changes between the measurement points F2, F4, F6, and F12 compared to the time of app completion (F0). Due to multiple testing with the same dependent variable (see outcomes and Figure 1), Bonferroni correction was applied, which led to the significance level being defined as *p* = 0.01 for all analysis using the Wilcoxon signed-rank test. Since the sample size of this study was smaller than expected, only one subgroup analysis was conducted, which distinguished between users with a low/moderate or high physical activity at baseline. According to the WHO, less than 3000 MET-minutes per week is defined as low to moderate, while a minimum of 3000 MET-minutes per week characterizes high levels of physical activity [42]. Furthermore, in order to identify factors influencing the effect of the intervention on physical activity and BMI, the Spearman’s Rho test was carried out as test for correlations as well as the Mann-Whitney-U test for independent samples. Due to the small sample size and the low significance, a regression test was omitted.

## 3. Results

### 3.1. Population

Between July 2019 and June 2020, 737 individuals registered for the “VIDEA bewegt” app program and were offered participation in the study. Of the 193 individuals who were interested in participating, 103 answered the first questionnaire. Of those, 90 study participants completed the program and the second questionnaire (F0). More than one quarter (25 of those 90) of potential study participants could not be included into the presented analysis for the following reasons: three people withdrew their study participation, eleven participants were excluded due to non-meaningful use (completion of more than 50% of the program in one day), another ten did not answer the first questionnaire (B0) before finishing the first two course weeks and/or the second questionnaire (F0) within one week after completion of the program and one person did not answer any of the follow-up questionnaires F2, F4, F6 and F12. Finally, 65 users could be included in this follow-up study with women accounting for 81.5% (*n* = 53) of the participants and a median participant age of 52 years (mean = 49 years, SD = 13.82). Most participants were married (58.5%, *n* = 38), had a university degree (38.5%, *n* = 25), and were insured through a statutory health insurance, which covered the course costs completely (83.1%, *n* = 54). The median duration of program use was 87 days (interquartile range (IQR) = 155.5 days). Further information on the characteristics of the participants is provided in Table 1.

### 3.2. Physical Activity

There were no significant differences in the primary outcome MET-minutes per week between B0 and any of the following measurement time points F0 (*p* = 0.150, r = 0.180), F2 (*p* = 0.242, r = 0.150), F4 (*p* = 0.515, r = 0.084), F6 (*p* = 0.766, r =0.040) and F12 (*p* = 0.032, r = 0.317) (see Table 2). Overall, study participants showed highest MET-minutes per week during F2 (median 3840 MET-minutes per week). By contrast, at F12 physical activity had significantly decreased by 960 MET-minutes per week (median) compared to the end of intervention F0 (*p* = 0.002, r = 0.462).

However, when comparing effects between participants with high and low/moderate baseline activity, a significant increase in physical activity in participants with a low or moderate activity at the beginning of the intervention was observed during the first four months after program completion (F0: *p* < 0.001, r = 0.684; F2: *p* < 0.001, r = 0.775; F4: *p* < 0.001, r = 0.649). In that period of time, the median MET-minutes per week increased by at least 140 MET-minutes per week at F0 and a maximum of 1200 MET-minutes per week at F2 compared to B0. F12 was the only measurement point where MET-minutes per week fell below the baseline level (−150 MET-minutes per week, *p* = 0.258, r = 0.222).

The physical activity of users with a high baseline activity showed an opposing trend, meaning that the median MET-minutes per week were below the baseline at all follow-up measurement points. At F2, physical activity was significantly below B0 (*p* = 0.009, r = 0.506) and at F12, the median physical activity had significantly decreased by 3640 MET-minutes per week compared to B0 (*p* < 0.001, r = 0.785) and by 3600 compared to F0 (*p* = 0.005, r = 0.646).

Nevertheless, the proportion of participants with a high physical activity level was always above baseline during the first six months of follow-up and only fell below B0 at F12 (see Table 2).

When comparing the sociodemographic characteristics of users depending on their baseline activity, it becomes clear that participants with a low to moderate baseline activity were significantly younger and had a higher level of education than users with a high baseline activity. It also took them less time to finish the program and they were more frequently informed about the intervention by their health insurance than people with a high baseline activity. Both groups of participants where predominantly employed full time, but while among participants with a heightened baseline activity were a higher proportion of retired people, users with a low or moderate baseline activity were more frequently unemployed (see Appendix C).

### 3.3. Health-Related Quality of Life

Health-related quality of life based on the Physical and Mental Component Summary Score did significantly improve between B0 and the measurement time points F0 (PCS: *p* = 0.003, r = 0.325; MCS: *p* < 0.001, r = 0.456) and F2 (PCS: *p* < 0.001, r = 0.521; MCS: *p* = 0.006, r = 0.344). Compared to baseline, the median PCS and MCS were increased at all measurement points except for F12, even though the changes at F4 and F6 did not meet the threshold of significance (see Table 3 and Table 4).

### 3.4. Additional Analyses

BMI was calculated from self-reported height and weight and was reduced continuously during the whole year of follow-up. Significant decreases were observed between F0 (*p* < 0.001, r = 0.441), F2 (*p* < 0.001, r = 0.478), F4 (*p* = 0.002, r = 0.409) and F6 (*p* < 0.001, r = 0.465) compared to B0 (see Table 5). Data show the strongest decrease in body weight for those participants with high body weight at baseline (see Appendix D).

Based on the GPAQ, the active minutes per day in the domains work, transport, and leisure time as well as the sedentary time per day were analysed. None of these outcomes showed any significant changes during follow-up compared to baseline (see Appendix E, Table A4). Nevertheless, the median of active minutes per day in the work domain increased during the first six months and even doubled at the measurement time points F2 and F4, while the other domains showed only minor improvements during the first months after the intervention.

The results of a Mann-Whitney-U test suggest that the development of the different domains was partly influenced by the season and by the presence or absence of high incidences of COVID-19 (see Appendix E, Table A4). At F0 (*p* = 0.020, r = 0.290) and F12 (*p* = 0.029, r = 0.322) participants showed a higher number of active minutes per day in the work domain during the warm season, as well as at F4 (*p* = 0.015, r = 0.325) and F12 (*p* = 0.020, r = 0.343) in the leisure time-domain. At F0 participants showed a higher number of active minutes per day in the transport domain (*p* = 0.015, r = 0.303) and in leisure time domain (*p* = 0.005, r = 0.355) during months with high incidences of COVID-19.

Furthermore, the sedentary time per day was decreased by one hour during the first 6 months after completion of the program (see Appendix E, Table A5).

### 3.5. User Assessment of the App Components

At the end of the intervention (F0), participants found it likely that they would continue using the app components “my focus” and “practical tips” as well as the in-app training sessions, the latter being the most favoured for a continuous use (see Figure 2).

During follow-up, a small minority reported performing the in-app training sessions or referring to the practical tips within the app daily, while the majority reported using both at least once a week, irregularly or never (see Figure 3 and Figure 4). The ratio between those using these two components at least once per week and those using them irregularly or never was stable over time, although a decrease in use could be observed.

In terms of sustained use of “my focus”, participants reported using predominantly the goal-setting (“my personal why”), motivational (“sources of strength”, “stop negative thoughts”) and support (“coping with setbacks”) components of the app (see Figure 5), although usage declined between F2 and F12. The components providing “rewards” and allowing for action planning (“action plan” and “if/then-plans”) were used less according to the participants.

A majority of users reported having improved their health literacy by using the app, a fact that remained stable throughout the whole follow-up period (see Figure 6).

For a detailed description of the intervention and the app components, please see Appendix A.

### 3.6. Effect Analyses

#### 3.6.1. Usage Time of the Intervention

There was no significant correlation between total usage time of the app and MET-minutes per week and BMI at most of the follow-up measurement points, but at F0 usage time (measured in days) was positively correlated with MET-minutes per week (r = 0.295, *p* = 0.018) (see Appendix F, Table A6). In contrast, at F4 a long usage time was significantly associated with a decline in physical activity compared to baseline (r = −0.270, *p* = 0.039) (see Appendix F, Table A7).

#### 3.6.2. Motivation for Continuous Use of App Components at F0

Motivation to further use the training videos correlated significantly with an increase in physical activity at F4 (r = 0.349, *p* = 0.006) and F6 (r = 0.291, *p* = 0.030) (see Appendix F, Table A6) and motivation to continue doing the exercises presented in the app component “my focus” positively correlated with an increase in MET-minutes per week at F0 (r = 0.258, *p* = 0.040), F4 (r = 0.390, *p* = 0.002), F6 (r = 0.356, *p* = 0.007) and F12 (r = 0.294, *p* = 0.047) (see Appendix F, Table A6). Furthermore, motivation to further use contents of “my focus” correlated with a decrease of BMI compared to B0 at F2 (r = −0.269, *p* = 0.033) and F4 (r = −0.277, *p* = 0.032) (see Appendix F, Table A9) and motivation to further use contents of “practical tips” also correlated with a decrease of BMI compared to B0 at F2 (r = −0.290, *p* = 0.021), F4 (r = −0.291, *p* = 0.024) and F6 (r = −0.266, *p* = 0.046) (see Appendix F, Table A9).

#### 3.6.3. Sustained Use

Continuous use of the in-app videos was positively correlated with physical activity at F2 (r = 0.279, *p* = 0.029), F4 (r = 0.390, *p* = 0.006), F6 (r = 0.528, *p* = 0.000) and F12 (r = 0.355, *p* = 0.015) (see Appendix F, Table A6) and also negatively corelated with BMI at F4 (r = −0.258, *p* = 0.048) and F6 (r = −0.447, *p* = 0.001) compared to baseline (see Appendix F, Table A9). Furthermore, there was a correlation between usage of practical tips and a decrease of BMI at F2 (r = −0.298, *p* = 0.018) and F6 (r = −0.359, *p* = 0.007) compared to B0 (see Appendix F, Table A9).

However, continuous use of the in-app videos also positively correlated with an increase in BMI at F12 (r = 0.452, *p* = 0.002) (see Appendix F, Table A8). 

Other factors such as the participation in other online or analogue lifestyle interventions or the external factors such as the COVID-19 pandemic did not seem to have any significant impact on the MET-minutes per week and the BMI of the study participants (see Appendix F, Table A10, Table A11, Table A12, Table A13).

## 4. Discussion

The results of the present study show that the app under investigation can actively increase the time spent physically active, especially for those with low levels of activity at baseline, and so it is in line with previous results concerning the effects of digital behavior change applications (e.g., for type 2 diabetes) [19].

With 81.5%, women accounted for the vast majority of study participants. This is an even higher proportion of female participants than in other studies [43,44] even though telemedicine interventions are known to be more likely used by women [12,45,46]. Still, the high proportion of women is representative for the users of “VIDEA bewegt” according to the manufacturer’s data.

With the majority of the participants being older than 50 (median age 52) and overweight or obese, it is evident that the intervention succeeded in reaching its target population. Furthermore, the results show the effectiveness of the app for those at high risk for cardiovascular events or lifestyle-associated diseases [47], an observation which is further underlined by the fact that significant effects on the MET-minutes per week were only to be found with those participants reporting low baseline levels. This effect is mirrored by the effects on weight loss, which were higher for overweight and obese participants. This corresponds with findings on improvement in clinical outcomes through the use of diabetes self-management apps [48].

However, the majority of participants is well-educated, underlining once again, that preventive measures, even though delivered via an app, often do not reach those especially vulnerable to life style-related diseases [49].

For most positive effects observed, there is a clearly visible wash-out period between F4 and F6, where values for physical activity decrease, in some cases drastically, and the BMI increases, suggesting that while the app may have had an activating effect [50], the impact reduces over time [11,51] if no new content is provided within the app [52]. 

Directly after finishing the app program, usage time positively correlated with physical activity, whereas at F4 a long usage time had a negative influence on physical activity. While the latter may seem counterintuitive at first glance, it can possibly be explained since in the case of “VIDEA bewegt”, a long usage time is predominantly caused by discontinued use as the program can be modified much more in its duration than in its intensity. Thus, it could be useful to increase the frequency of reminders to guarantee a continuous and therefore meaningful use. At F12, values for MET-minutes per week drop below baseline values. This points at well-documented social desirability effects [53] which might have been present when estimating the baseline activity levels. However, it also suggests using the app might have served as a reality check concerning everyday activity levels for some participants [54]. This awareness-raising effect of the app [55] is further underlined by the fact that especially goal-setting components were used continuously throughout the whole follow-up period. While goal-setting is the basis for behavior change, keeping up with the behavior changes requires making plans to integrate it into everyday life [20]. The fact that the planning components of the app were used less according to the participants may be one explanation for the deteriorating effects on MET-minutes per week and the time spent active during work, transportation, and leisure time. The reported interest in the guided exercises within the app, which partly correlates with MET-minutes per week, fits the evidence for the effectiveness of in-app videos for promoting physical activity [56].

Quality of life, even though not always significant, remained high all the way to F6, which shows that quality of life corresponds with increased physical activity and decreases (e.g., in BMI) [57]. It also fits the reported high usage of app-components that provided motivational support, which shows the importance of positive framing of behavior change in order to fulfill the users’ outcome expectancy [58], a fact already known from interventions aiming to increase screening behavior [59]. The fact that quality of life also dropped below baseline at F12 once more shows the importance of continuously proving new content, also to prevent disinterest on the part of the users [60].

It is also worth noting that the median sedentary time of study participants decreased by one hour during the first six months of follow-up. Although this observation did not meet the threshold of significance, it is still of clinical relevance since one hour spent physically active instead of sitting already reduces all-cause mortality [61].

It is known that physical activity is influenced by external factors such as the season [62,63]. It is therefore not surprising that participants showed higher levels of physical activity at several measurement time points during the warm season. Furthermore, directly after finishing the app, participants spent more minutes per day physically active during time periods with high COVID-19 incidences. However, since these effects only occurred sporadically and may have been influenced by other external factors, no clear interpretation of this observation is possible.

Last but not least, decreases of weight and BMI peak at F6 while increases of MET-minutes and quality of life peak at F2 or F4, showing that actual metabolic effects of a digital intervention take time to show [52], warranting, once more, longer follow-ups [64] for evaluation studies.

### 4.1. Strengths and Weaknesses

The greatest strength of the present study lies in the reliance on real-world evidence, since data was generated in the every-day settings of participants [65]. Then again, the one-armed observational design holds considerable risk of bias, which has to be considered when interpreting the results. The lack of a control group and the unobserved app usage as well as completion of the questionnaires severely limits options to control for confounding factors. However, use of other health applications or courses was part of the questionnaires and had no significant effect on any of the outcomes studied. Still, showing correlations between app components and, respectively, BMI, weight and MET-minutes per week during app usage and follow-up periods, the study serves as a proof-of-concept for the usefulness of health apps with a video component for the promotion of physical activity.

The low number of participants, especially at F12, did not allow for more sophisticated analyses such as regression, rendering correlation results open for interpretation concerning the direction of the effects. This is also due to the fact that no power analysis or intention-to-treat calculation was performed a priori for this follow-up study. However, a power analysis was conducted concerning the complete evaluation of “VIDEA bewegt” (short-term and long-term study), which led to a study sample of at least 27 participants [24]. Even though the number of participants meets the required minimum, the sample size is far below the required number for subgroup analyses. This was caused by a low number of app users rather than a lack of willingness to participate in the study. Consequently, the intervention “VIDEA bewegt” encountered two of the leading barriers of telemedicine interventions to their implementation in medical care, which are non-use and discontinued use, mostly caused by user-related factors [66]. The attrition rate of this follow-up study was rather low. However, the previous study investigating the short-term effect of the intervention reported a significant dropout rate. Only 63% of the initial study participants took part in this follow-up study. Thirteen participants dropped out of the intervention, mostly at the beginning of the program, and another 25 participants were excluded since they did not fulfil the inclusion criteria. Thus, the population of this long-term evaluation is heavily filtered with a consecutive risk of selection bias. During the follow-up study the response rate was 100% at F0, 94% at F4, 95% at F6, and 92% at F12.

### 4.2. Outlook

Data collection of this study will carry on until January of 2022. Further research should apply “VIDEA bewegt” in a controlled setting without abandoning the real-world approach taken so far. The findings from the current proof-of-concept should be validated using at least a randomized control group and assign participants to both groups based on a power estimation.

Continuous evaluation of “VIDEA bewegt” should be performed especially when the app content is updated.

## 5. Conclusions

This follow-up study provides evidence on the positive medium-term effect of the intervention “VIDEA bewegt” on several clinical outcomes such as physical activity, health related quality of life and BMI, when being used by individuals with low levels of physical activity in everyday life. As such, it is an example for the potential of evidence-based video-interventions and contributes to a research field in which limited evidence exists to date. However, its validity is limited due to a small sample size due to recruitment problems and high dropout rates. 

## Figures and Tables

**Figure 1 nutrients-13-04215-f001:**
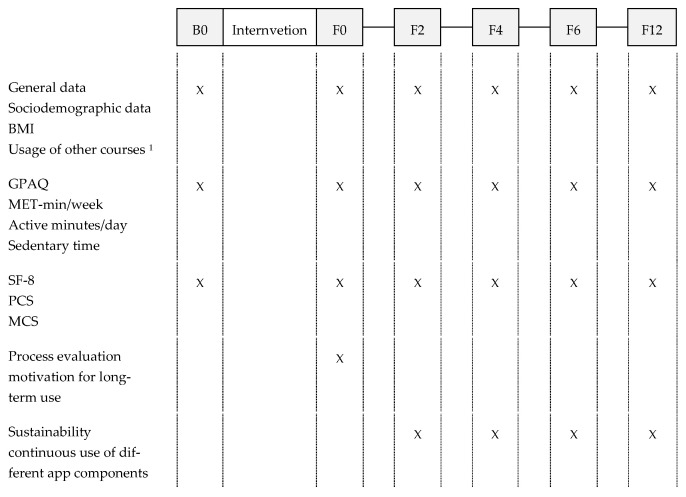
Study concept: measurement points and methods. ^1^ other analogue or digital lifestyle interventions than “VIDEA bewegt”. Abbreviations: B0 = baseline; F0 = end of intervention; F2 = two months after the program; F4 = four months after the program; F6 = six months after the program; F12 = twelve months after the program; GPAQ = Global Physical Activity Questionnaire, MET = metabolic equivalent, SF-8 = Short Form Health Survey; PCS = Physical Component Summary Score; MCS = Mental Component Summary Score.

**Figure 2 nutrients-13-04215-f002:**
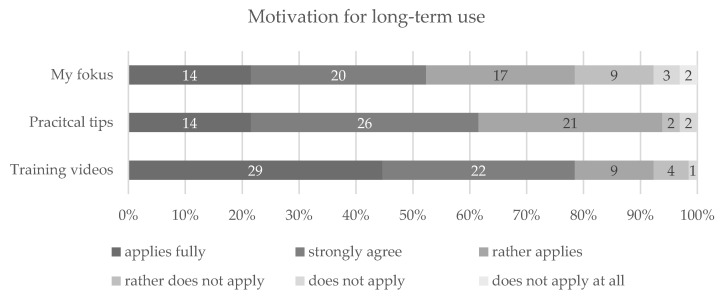
Motivation for long-term usage of app components at the end of the intervention. Notes: presentation in percentages.

**Figure 3 nutrients-13-04215-f003:**
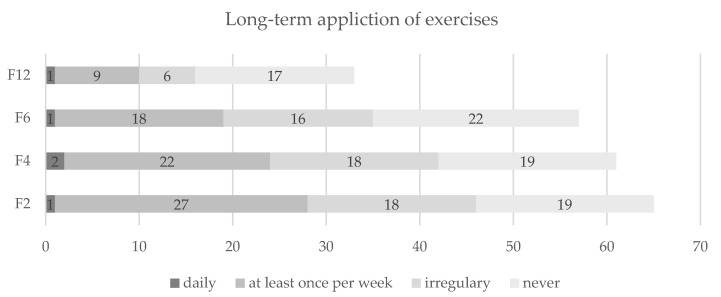
Sustained use of exercises contained in the training videos. Notes: Presentation in numbers of participants. Abbreviations: F2 = two months after the intervention; F4 = four months after the intervention; F6 = six months after the intervention; F12 = twelve months after the intervention.

**Figure 4 nutrients-13-04215-f004:**
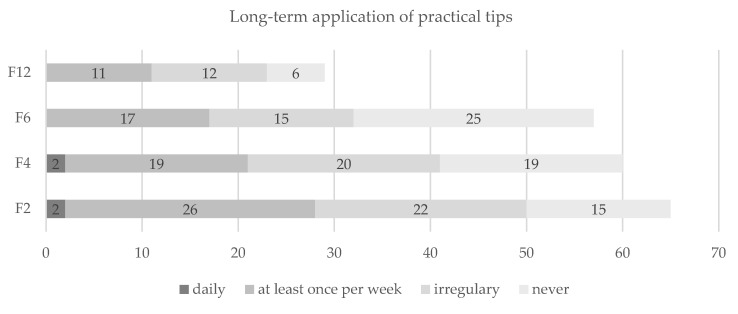
Sustained use of practical tips contained in videos of the app. Notes: Presentation in numbers of participants. Abbreviations: F2 = two months after the intervention; F4 = four months after the intervention; F6 = six months after the intervention; F12 = twelve months after the intervention.

**Figure 5 nutrients-13-04215-f005:**
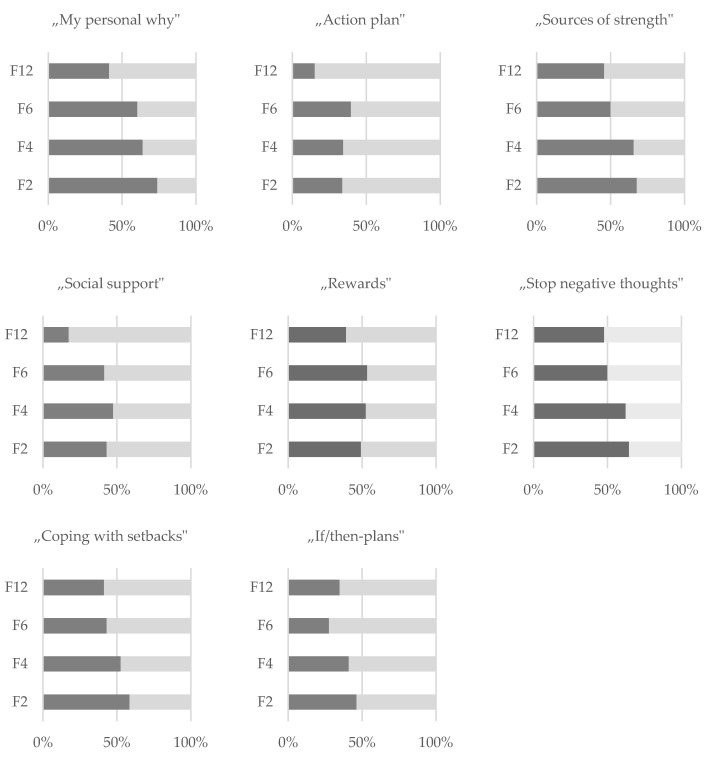
Application of different components of the rubric “my focus”. Abbreviations: F2 = two months after the intervention; F4 = four months after the intervention; F6 = six months after the intervention; F12 = twelve months after the intervention.

**Figure 6 nutrients-13-04215-f006:**
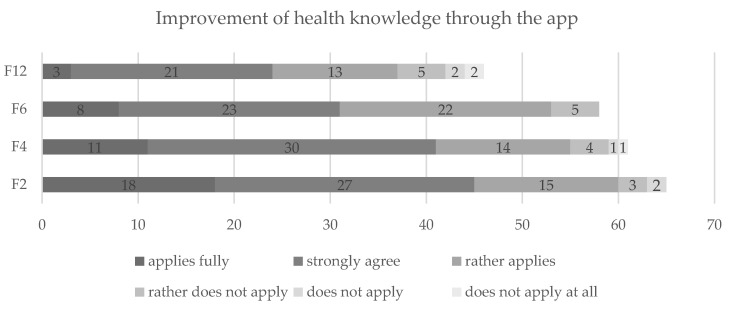
Long-term improvement of health knowledge due to “VIDEA bewegt”. Notes: Presentation in numbers of participants. Abbreviations: F2 = two months after the intervention, F4 = four months after the intervention, F6 = six months after the intervention, F12 = twelve months after the intervention.

**Table 1 nutrients-13-04215-t001:** Characteristics of the study participants.

Overall *n*	65	
**Sex [%, (*n*)]**		
female	81.5	(53)
male	18.5	(12)
**Age [years]**		
Mean, Median	49	52
20–29 [%, (*n*)]	12.3	(8)
30–39 [%, (*n*)]	13.8	(9)
40–49 [%, (*n*)]	18.5	(12)
50–59 [%, (*n*)]	29.2	(19)
60–69 [%, (*n*)]	21.5	(14)
70–79 [%, (*n*)]	4.6	(3)
**BMI**		
Mean, Median [kg/m^2^]	27.04	26.11
Normal weight [%, (*n*)]	45.3	(29)
Overweight [%, (*n*)]	25	(16)
Obesity [%, (*n*)]	29.7	(19)
**Marital status [%, (*n*)]**		
Married	58.5	(38)
Living in a stable relationship	13.8	(9)
Divorced or separated	12.3	(8)
Single	10.8	(7)
Widowed	3.1	(2)
Other	1.5	(1)
**Level of education [%, (*n*)]**		
University degree	38.5	(25)
Completed vocational training	32.3	(21)
High school (12 years or more)	10.8	(7)
Secondary school (10 or 11 years)	13.8	(9)
Main school (9 years or less)	1.5	(1)
Other	3.1	(2)
**Employment status [%, (*n*)]**		
Full-time	46.2	(30)
Half-time	15.4	(10)
Part-time	7.7	(5)
Not employed	10.8	(7)
Retired	20	(13)
**Length of program use [days]**		
Mean, Median	136.7	87
**Source of information ^1^ [%, (*n*)]**		
Health insurance	53.8	(35)
Doctor	15.4	(10)
Internet	15.4	(10)
Social environment	13.9	(9)
Others	1.5	(1)
**Participation in other sport courses ^2^ [%, (*n*)]**	46.2	(30)
**Use of other health apps ^2^ [%, (*n*)]**	21.5	(14)
**Health insurance provider [%, (*n*)]**		
AOK PLUS, AOK Rheinland/Hamburg ^3^	83.1	(54)
Other statutory insurance ^4^	13.8	(9)
Private insurance ^4^	3.1	(2)

^1^ about the app “VIDEA bewegt”; ^2^ interventions other than “VIDEA bewegt”; ^3^ German statutory insurances where users could participate in the app for free; ^4^ other German insurances where users had to pay 130€ in advance and were reimbursed part of the course cost after completing the program.

**Table 2 nutrients-13-04215-t002:** Development of MET-minutes per week, using Wilcoxon signed-rank test.

		B0	F0	F2	F4	F6	F12
all participants	*n*	64	64	61	60	56	46
	Mean	4680	5689	5090	4908	4876	3224
	(SD)	(4938)	(6903)	(4933)	(4687)	(4613)	(3687)
	Median	2640	3080	3840	3130	3560	2120
	25% quantile	1470	1440	1720	1280	1070	840
	75% quantile	6650	7440	6960	6570	8372	3630
	Wilcoxon ^1^						
	*p*		0.150	0.242	0.515	0.766	0.032
	z		−1.441	−1.171	−0.650	−0.297	−2.147
	r		0.180	0.150	0.084	0.040	0.317
	Wilcoxon ^2^						
	*p*			0.448	0.642	0.261	0.002
	z			−0.759	−0.465	−1.124	−3.136
	r			0.097	0.060	0.150	0.462
<3000 MET−min/week at B0	*n*	34	33	33	32	30	26
	Mean	1435	3141	3538	3084	2853	2005
	(SD)	(799)	(4112)	(3439)	(3204)	(3017)	(1753)
	Median	1560	1700	2760	1780	1730	1410
	25% quantile	820	980	1080	1200	855	780
	75% quantile	2040	3400	4540	3840	4005	3250
	Wilcoxon ^1^						
	*p*		<0.001	<0.001	<0.001	0.018	0.258
	z		−3.931	−4.450	−3.674	−2.368	−1.130
	r		0.684	0.775	0.649	0.432	0.222
	Wilcoxon ^2^						
	*p*			0.352	0.902	0.432	0.124
	z			−0.931	−0.123	−0.786	−1.537
	r			0.162	0.022	0.144	0.301
≥3000 MET−min/week at B0	*n*	30	30	27	27	25	19
	Mean	8357	8640	7159	7233	7461	5022
	(SD)	(5091)	(8233)	(5770)	(5212)	(5006)	(4881)
	Median	7060	7020	5180	5280	5280	3420
	25% quantile	4390	3000	3360	3220	3572	2020
	75% quantile	10,670	10,725	9240	10,080	11,370	7800
	Wilcoxon ^1^						
	*p*		0.565	0.009	0.110	0.211	<0.001
	z		−0.576	−2.631	−1.598	−1.251	−3.421
	r		0.105	0.506	0.308	0.250	0.785
	Wilcoxon ^2^						
	*p*			0.088	0.692	0.264	0.005
	z			−1.706	−0.396	−1.117	−2.817
	r			0.328	0.076	0.223	0.646
<3000 MET−min/week [% (*n*)]		53.1 (34)	48.4 (31)	36.1 (22)	45.0 (27)	46.4 (26)	58.7 (27)
≥3000 MET−min/week [% (*n*)]		46.9 (30)	51.6 (33)	63.9 (39)	55.0 (33)	53.6 (30)	41.3 (19)

^1^ asymptotic, two-sided Wilcoxon test between the baseline and subsequent measurement points; ^2^ asymptotic, two-sided Wilcoxon test between the end of intervention (F0) and the subsequent measurement points. Abbreviations: B0 = baseline; F0 = end of intervention; F2 = two months after the intervention; F4 = four months after the intervention; F6 = six months after the intervention; F12 = twelve months after the intervention; MET = metabolic equivalent; r = effect size.

**Table 3 nutrients-13-04215-t003:** Development of PCS, using Wilcoxon signed-rank test.

	B0	F0	F2	F4	F6	F12
*n*	65	65	65	61	58	45
Mean	47.24	50	51.41	48.71	49	48.31
(SD)	(8.83)	(6.94)	(6.71)	(9.04)	(8.05)	(9.03)
Median	49.74	50.48	52.34	51.11	50.16	48.92
25% quantile	41.30	44.61	47.76	45.48	42.04	42.10
75% quantile	53.30	55.46	56.68	55.89	55.86	56.96
Wilcoxon ^1^						
*p*		0.003	<0.001	0.033	0.025	0.167
z		−2.623	−4.197	−2.133	−2.241	−1.383
r		0.325	0.521	0.273	0.294	0.206

^1^ Wilcoxon signed-rank test between B0 and the follow-up measurements points. Abbreviations: B0 = baseline; F0 = end of intervention; F2 = two months after the intervention; F4 = four months after the intervention; F6 = six months after the intervention; F12 = twelve months after the intervention; PCS = Physical Component Summary Score; r = effect size.

**Table 4 nutrients-13-04215-t004:** Development of MCS, using Wilcoxon signed-rank test.

	B0	F0	F2	F4	F6	F12
*n*	65	65	65	61	58	45
Mean	47.22	51.59	50.28	49.25	48.78	45.4
(SD)	(9.05)	(8.23)	(7.86)	(10.02)	(9.74)	(10.04)
Median	48.30	52.55	51.10	52.31	49.87	46.70
25% quantile	41.57	48.01	47.88	44.01	43.42	40.35
75% quantile	53.50	57.48	56.88	57.48	57.10	51.98
Wilcoxon ^1^						
*p*		<0.001	0.006	0.054	0.027	0.800
z		−3.676	−2.773	−1.952	−2.210	−0.254
r		0.456	0.344	0.250	0.290	0.038

^1^ Wilcoxon signed-rank test between B0 and the follow-up measurements points. Abbreviations: B0 = baseline; F0 = end of intervention; F2 = two months after the intervention; F4 = four months after the intervention; F6 = six months after the intervention; F12 = twelve months after the intervention; MCS = Mental Component Summary Score; r = effect size.

**Table 5 nutrients-13-04215-t005:** Development of the BMI, using Wilcoxon signed-rank test.

	B0	F0	F2	F4	F6	F12
*n*	64	64	63	60	57	46
Mean	27.04 (5.61)	26.57 (5.38)	26.56 (5.17)	26.45 (5.25)	26.31 (5.01)	26.06 (4.92)
(SD)						
Median	26.12	24.91	25.38	25.44	25.16	24.85
25% quantile	22.55	22.52	22.59	22.16	22.50	24.85
75% quantile	31.22	30.59	30.10	29.75	29.34	30.12
Wilcoxon ^1^						
*p*		<0.001	<0.001	0.002	<0.001	0.624
z		−3.530	−3.794	−3.169	−3.514	−0.490
r		0.441	0.478	0.409	0.465	0.072

^1^ Wilcoxon signed-rank test between B0 and the follow-up measurements points. Abbreviations: B0 = baseline; F0 = end of intervention; F2 = two months after the intervention; F4 = four months after the intervention; F6 = six months after the intervention, F12 = twelve months after the intervention; SD = standard deviation; r = effect size.

## Data Availability

Data will be shared with researchers who provide a methodologically sound proposal. Proposals should be directed to peter.schwarz@uniklinikum-dresden.de. To gain access, data requestors will need to sign a data access agreement.

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
