# Peer review of "Long-Term Effects of a Video-Based Smartphone App (“VIDEA Bewegt”) to Increase the Physical Activity of German Adults: A Single-Armed Observational Follow-Up Study"

_nutrients, 2021, doi:10.3390/nu13124215_

Round 1
Reviewer 1 Report
2.2. Intervention
All the behavioral intervention elements of the intervention needs to be discussed and detailed and cited to demonstrate an evidenced based approach.
Simply stating ” The app “VIDEA bewegt” is a certified, digital preventive program which aims to increase the physical activity of its users in everyday life by videos combining educational content and training instructions. Furthermore, the app contains several additional components to develop and maintain motivation and behaviour change such as goal setting, documentation of progress and personal messages.”
Why would this be considered novel or effective ?
2.4. Primary and secondary outcomes
1. The authors need to discuss the validity of the tools used as related to this population.
2. The authors state, “Further outcomes included the BMI and bodyweight, physically active minutes per week in the domains of leisure, work, and transport, as well as sedentary hours per day.”
Is this all-self-report?
3. All measures and cut points need to be clearly defined and cited in Methods.
N= 65 is a very small sample for this type of study and does not provide much meaning. All though nonparametric statics were used, the statistical power should be reported.
Table 1 should be redone to simply show the mean/median characteristics of the sample.
Using quantiles or age distribution for a very small population is not appropriate.
There are too many tables included in the main manuscript. Only demographics and one or two main results tables should be presented.
There are no novel findings presented in this study.
Reviewer 2 Report
The manuscript describes the impact of using the app on motivating people to increase their physical activity. This aspect is very important as due to growing number of patients with obesity, diabetes and other sedentary-related lifestyle diseases.
Here are my minor remarks:
Abstract
„medium 52 years” - mean, median?
In abstract it is expect to give values (for instance mean or average) and their change. The p-value is not necessary - only information whether the change was significant or not. The MCS, PCS, and BMI values were not given – this should be corrected.
Methods
page 4 - „bodyweight” – was used alone, but maybe “bodyweight exercise” should be used
Authors decided to use a questionnaire-based MET-minutes parameter, but VO2max could be a better alternative to assess change in participants’ fitness.
Results
The most numerous groups of participants were those of age 50 – 59 (29.2%) and 60 – 69 (21.5%). Younger and older were less numerous. This could have an impact on all comparisons where age is important.
“Table 1. Characteristics of the study participants” – this should be placed in the “Methods/Participants” section as this is not result though taken from questionnaires.
“r” is first explained under Table 10, not at first use.
Discussion
“The results of the present study show that apps for …” – in the study you tested only one app, thus the results cannot be used for making conclusions on some other apps.
If Ref. no. 24 is not available, should it be referenced in this manuscript?
“on reductions in clinical outcomes through the use of diabetes self-management apps [39]” this suggests that the apps had a negative influence. Is this correct?
“13 participants dropped out of the intervention” – the number at the beginning of the sentence must be provided in writing.
